# The phospho-docking protein 14-3-3 regulates microtubule-associated proteins in oocytes including the chromosomal passenger Borealin

**Charlotte Repton**[1], **C. Fiona Cullen**[1], **Mariana F. A. Costa**[1], **Christos Spanos**[1],
**Juri Rappsilber**[1,2], **Hiroyuki Ohkura**[1]*

**1** Wellcome Centre for Cell Biology, School of Biological Sciences, University of Edinburgh, Edinburgh, Scotland, United Kingdom, **2** Chair of Bioanalytics, Institute of Biotechnology, Technische Universität Berlin, Berlin, Germany

* h.ohkura@ed.ac.uk

## Abstract

Global regulation of spindle-associated proteins is crucial in oocytes due to the absence of centrosomes and their very large cytoplasmic volume, but little is known about how this is achieved beyond involvement of the Ran-importin pathway. We previously uncovered a novel regulatory mechanism in *Drosophila* oocytes, in which the phospho-docking protein 14-3-3 suppresses microtubule binding of Kinesin-14/Ncd away from chromosomes. Here we report systematic identification of microtubule-associated proteins regulated by 14-3-3 from *Drosophila* oocytes. Proteins from ovary extract were co-sedimented with microtubules in the presence or absence of a 14-3-3 inhibitor. Through quantitative mass-spectrometry, we identified proteins or complexes whose ability to bind microtubules is suppressed by 14-3-3, including the chromosomal passenger complex (CPC), the centralspindlin complex and Kinesin-14/Ncd. We showed that 14-3-3 binds to the disordered region of Borealin, and this binding is regulated differentially by two phosphorylations on Borealin. Mutations at these two phospho-sites compromised normal Borealin localisation and centromere bi-orientation in oocytes, showing that phospho-regulation of 14-3-3 binding is important for Borealin localisation and function.

## Author summary

Accurate segregation of chromosomes during cell division is fundamental for genome stability.

Chromosome segregation is mediated by the spindle, which is made of dynamic microtubules and associated proteins that regulate microtubule behaviour. How these microtubule-associated proteins are regulated is not well understood. Furthermore, as oocytes have an exceptionally large volume of cytoplasm and lack centrosomes, regulation of microtubule-associated proteins is especially crucial for organisation and function of the

**Data Availability Statement:** Raw mass spectrometry proteomics data were deposited to the ProteomeXchange Consortium (http://

proteomecentral.proteomexchange.org) via the PRIDE partner repository with the dataset identifier PXD030445.

**Funding:** This work is supported by grants from the Wellcome Trust (wellcome.org; 098030, 206315) and Biotechnology and Biological Sciences Research Council (BBSRC; bbsrc.ukri. org; BB/S013059) to H.O. C.R. received a PhD studentship from BBSRC EastBio Doctoral Training Partnership. Wellcome Centre for Cell Biology is supported by core funding from the Wellcome Trust (092076 and 203149). The funders had no role in study design, data collection and analysis, decision to publish, or preparation of the manuscript.

**Competing interests:** The authors have declared that no competing interests exist.

meiotic spindle. In this study, we showed that 14-3-3, a protein that binds to phosphorylated proteins, plays an important role to regulate multiple microtubule-associated proteins in fly oocytes. The regulated proteins include subunits of the conserved kinase complex called the chromosomal passenger complex. We further found that interaction of one of the subunits with 14-3-3 is regulated by two phosphorylations, and that these phosphorylations are important for localisation and function of this subunit. As these proteins are widely conserved, including in humans, this study may provide an insight into chromosome mis-segregation in human oocytes, which is very frequent and a major cause of human infertility, miscarriages and congenital birth conditions.

## Introduction

Oocytes are specialised cells that undergo meiotic divisions to produce female gametes. Oocytes form the meiotic spindle to segregate chromosomes using mechanisms shared with mitotic cells, but also need to cope with unique challenges to assemble a spindle. The centrosomes are absent during meiotic spindle assembly in the oocytes of most animals, such as humans and *Drosophila melanogaster* [1], even though the centrosomes are the canonical microtubule organising centres in mitosis. Furthermore, oocytes have an exceptionally large volume of cytoplasm, but need to limit spindle assembly to only the region around the chromosomes and not in other parts of the oocytes. A failure to suppress spindle assembly away from the chromosomes would lead to inefficient use of cellular resources, and most likely lead to aberrant cellular processes. Therefore, crucial proteins for spindle assembly are likely to be activated only around chromosomes but inactivated away from chromosomes.

The spatial regulation of spindle assembly in oocytes is not yet well understood, but one mechanism involving Ran and importin has been well described. As first shown in *Xenopus* egg extract, chromosome-associated RanGEF, Rcc1, converts Ran-GDP to Ran-GTP. Then Ran-GTP releases spindle assembly factors, such as TPX2, from inhibitory effects of importins to allow microtubule assembly around chromosomes (reviewed in [2]).

In living oocytes, there is mixed evidence for a role for Ran in chromosome-dependent spindle formation. On one hand, there is evidence that expression of dominant negative Ran impairs or delays meiosis I spindle assembly in human and mouse oocytes [3,4]. On the other hand, even when the Ran gradient was disrupted in a mouse oocyte by expressing hyperactive or dominant negative Ran, the oocyte still assembled a single spindle around chromosomes in meiosis I [5]. Similarly, in *Drosophila* oocytes, expression of dominant-negative or hyperactive Ran induced mild spindle defects in meiosis I but did not produce ectopic spindles [6]. This suggests that oocytes may have other pathways to spatially regulate spindle assembly, in addition to the Ran-importin pathway. The chromosomal passenger complex (CPC), containing Aurora B kinase, was proposed to provide an alternative chromosomal signal independently of Ran [7,8].

Recently, we proposed a new spatial regulation mechanism in *Drosophila* oocytes, involving the phospho-docking protein 14-3-3. 14-3-3 proteins are a well-conserved family of small, ubiquitous phospho-docking proteins involved in various cellular processes [9]. The 14-3-3 family is conserved across eukaryotes, with at least 7 different isoforms present in the human genome, while yeast, *Caenorhabditis elegans* and *Drosophila melanogaster* have 2 each (reviewed in [10,11]). 14-3-3 primarily exists as a dimer, either a homodimer of the same isoform or a heterodimer of two different isoforms.

Most of the described roles of 14-3-3 are through its direct binding to a target protein, with over 200 binding sites reported in the literature [12]. The majority of reported sites are centred on a phosphorylated serine or threonine, while in some rare cases 14-3-3 can bind in the absence of phosphorylation [13]. There are multiple mechanisms through which 14-3-3 can regulate its targets. For example, 14-3-3 can mask localisation motifs to regulate sequestration or shuttling of targets in or between subcellular compartments [14,15]. Some cases have also been reported of the 14-3-3 dimer acting as an adaptor, bringing together two proteins to increase their activity or interaction [16]. Additionally, Yaffe et al [17] propose that 14-3-3 might act as a "molecular anvil" to instigate conformational change in its binding partners. Finally, 14-3-3 might act as a chaperone of unfolded proteins, or facilitate clustering of proteins or complexes.

The importance of 14-3-3 in spindle formation in oocytes has already been demonstrated. 14-3-3 knockdown in *Drosophila* oocytes leads to defects in spindle bipolarity [18]. Of the two *Drosophila* isoforms, knockdown of one (14-3-3ε) alone is sufficient to cause these spindle defects in oocytes. Although knockdown of the other isoform (ζ) alone showed no defects, double knockdown with 14-3-3ε resulted in much more severe spindle defects than 14-3-3ε knockdown alone. Similarly to *Drosophila*, depletion of 14-3-3η in mouse oocytes disrupts spindle assembly and polar body extrusion [19]. We previously identified a microtubule-cross-linking kinesin Ncd as a 14-3-3 target critical for spindle bipolarity in *Drosophila* oocytes [18]. Ncd and the mammalian orthologue HSET belong to the kinesin-14 family of minus-end directed microtubule motors [20], and both are required to focus the poles of the spindle in oocytes [21,22]. Our previous study proposed the mechanism by which 14-3-3 and Aurora B kinase allow Kinesin-14/Ncd binding of microtubules only near the chromosomes [18]. In the ooplasm away from chromosomes, 14-3-3 binds to Kinesin-14/Ncd at phosphorylated Serine 96, preventing Kinesin-14/Ncd from binding microtubules. In the vicinity of the chromatin, an additional phosphorylation nearby at Serine 94, by the chromatin-bound kinase Aurora B, inhibits 14-3-3 binding of kinesin-14/Ncd. This removes the inhibition, and specifically allows Kinesin-14/Ncd binding to spindle microtubules, promoting proper bipolar spindle formation.

Given the importance of spatial regulation for the oocyte, we ask if 14-3-3 regulates other important spindle-associated proteins in oocytes, and what mechanisms might be involved. By developing a new biochemical method, we identify various microtubule-associated proteins regulated by 14-3-3 in ovary extract. Among these proteins, of particular interest is the chromosomal passenger complex (CPC) subunit Borealin, because the CPC is one of the master regulators of cell division and not known to be regulated by 14-3-3 in any system. We show that Borealin is bound by 14-3-3, and this binding is differentially regulated by two phosphorylations. Mutations in these phospho-sites disrupt Borealin localisation to the spindle equator/centromeres and function in centromere bi-orientation. Collectively, these findings suggest that 14-3-3 plays a central role in a general regulatory system controlling microtubule-associated proteins (MAPs) in oocytes, analogous to the Ran-importin system.

## Results

### Identification of proteins regulated by 14-3-3 from *Drosophila* oocytes

We recently proposed a new mechanism of spatial regulation in oocytes, in which Kinesin-14/Ncd is regulated by the combined action of the phospho-docking protein 14-3-3 in the ooplasm and Aurora B kinase on chromosomes [18].

We hypothesise that this mechanism involving 14-3-3 and Aurora B provides a general way to spatially activate many spindle proteins only when near the chromosomes in oocytes. Here

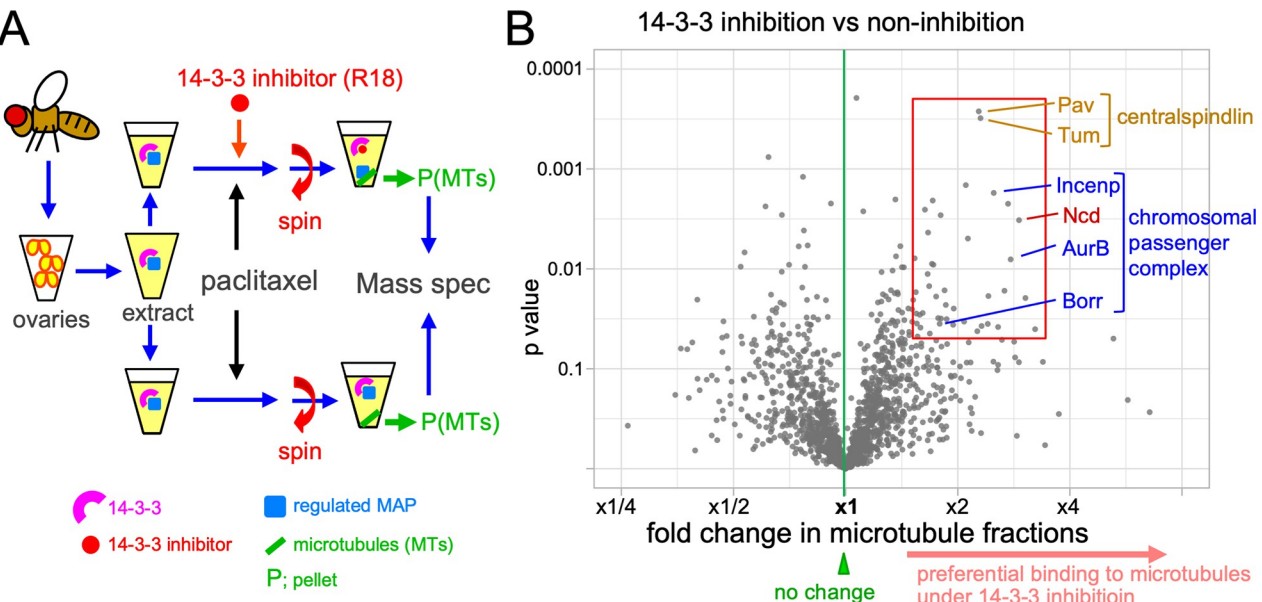

**Fig 1. Identification of proteins whose microtubule binding is regulated by 14-3-3.** (A) Dissected ovaries were homogenised and cleared by centrifugation to make soluble extract. It was separated into two and a 14-3-3 inhibitor (R18) was added to one. After microtubules are polymerised by addition of Taxol, they are sedimented by centrifugation through sucrose cushion. The pellets containing microtubules and their associated proteins were analysed by label-free quantitative mass-spectrometry. (B) Volcano plot showing the fold changes of the amounts of each protein detected in microtubule fraction in the presence of the 14-3-3 inhibitor in comparison to its absence on the X axis and the significance (p-value) on the Y axis. The red box contains 47 proteins that significantly increased their microtubule binding under 14-3-3 inhibition (the fold change > . 1.5 and the p < 0.05).

we tested this hypothesis by developing a new biochemical method to systematically identify proteins whose ability to bind microtubules is regulated by 14-3-3 in oocytes.

First, we have developed a novel method of purifying microtubule-associated proteins (MAPs) from *Drosophila* ovaries (**Fig 1A**) by adapting a previous method for embryos [23]. We took advantage of *Drosophila* ovaries being mainly made up of mature oocytes (arrested in metaphase I) by volume with a minimal contribution from mitotic cells. A hundred pairs of ovaries were dissected from mature flies and homogenised on ice to depolymerise microtubules. After the resulting lysate was cleared by ultracentrifugation, endogenous tubulin was polymerised by incubating with paclitaxel and GTP at room temperature. Microtubules, together with their associated proteins, were pelleted through a sucrose cushion by ultracentrifugation. The pellet predominantly consists of tubulins and minimal amounts of other proteins (**S1 Fig**)

Next we used a 14-3-3 inhibitor, R18. R18 is a synthetic peptide with 20 residues that has been shown to competitively bind to the 14-3-3 phospho-docking site with a higher affinity than native substrates [24]. This inhibitor has been shown to bind to various 14-3-3 isoforms, including ε and ζ in vertebrates [24–26]. We expect it can inhibit both 14-3-3 isoforms (ε and ζ) in *Drosophila* oocytes, as the same isoforms in *Drosophila* and vertebrates are more similar to each other than between two different isoforms in the same species. To identify MAPs regulated by 14-3-3 in oocytes, soluble ovary extract from wild type was divided into two aliquots. The 14-3-3 inhibitor R18 was added to one, and as a control, water was added to the other. After incubation with paclitaxel and GTP, microtubules and associated proteins were pelleted from both samples.

These two microtubule fractions were then analysed by LC-MS/MS for identification and label-free quantification.

## Quantitative mass-spectrometry has identified proteins potentially regulated by 14-3-3

To determine the relative amounts of each protein in the microtubule fraction with or without 14-3-3 inhibition, six pairs of biological replicates were analysed by label-free quantification using LC-MS/MS (S2–S7 Tables). As a result, we have detected 3,564 proteins in ovaries, including 24 out of 32 known spindle MAPs in our dataset. Among them, we were able to quantify 1,504 proteins (42%) in at least 2 out of the 6 experiments (S1 Table). These proteins were plotted as a volcano plot with fold differences on the X axis and the confidence level (p-value) on the Y axis (Fig 1B and S1 Table). In this plot, proteins on the top right increased their binding to microtubules under 14-3-3 inhibition. In total, 47 proteins were detected in microtubule fractions at a higher abundance in the presence of the 14-3-3 inhibitor than in its absence with a good confidence (defined by $p < 0.05$ and ratio $> 1.5$; Fig 1B). 14-3-3 is likely to suppress microtubule binding of these proteins in oocytes. Using equivalent criteria, 20 proteins were detected in microtubule fractions at a lower abundance in the presence of the 14-3-3 inhibitor than in its absence (S2 Fig).

It is possible that some identified proteins are not regulated directly by 14-3-3, and instead form a complex with a protein regulated by 14-3-3. To test this possibility, we examined whether any of these 14-3-3 regulated proteins or their orthologues are known to physically interact with each other using STRING database ([27]; Figs 2A and S2). Among the 47 proteins that increased microtubule binding in the presence of the 14-3-3 inhibitor, we found 48 known protein-protein interactions, which is significantly higher than expected from a random set of 47 proteins (17 interactions; $p = 1.2e^{-09}$). This is consistent with the possibility that microtubule binding of some proteins is indirectly regulated by 14-3-3 through interactions with other proteins.

One of the proteins that increased binding to microtubules under 14-3-3 inhibition with a high degree and confidence was a known 14-3-3 regulated protein in oocytes, Kinesin-14/Ncd (Fig 1B). In addition, both subunits of centralspindlin (MKlp1/Pav, RacGAP/Tum) increased their binding to microtubules under 14-3-3 inhibition (Figs 1B and 2A). This is consistent with a previous study that shows MKlp1/Pav is directly regulated by 14-3-3 in human mitotic cells [28], although this has not been shown in oocytes. These results confirmed that our new method can successfully identify 14-3-3 regulated proteins. Similarly, three out of the four CPC subunits (Aurora B, Incenp and Borealin) significantly increased microtubule binding under 14-3-3 inhibition, except the smallest subunit Survivin/Deterin (Figs 1B and 2A). The CPC is not yet known to be regulated by 14-3-3 in any organism, and therefore was investigated further.

A broader question is whether 14-3-3 regulates only a few microtubule-associated proteins or many proteins in oocytes. To gain an insight into this question, we tested whether the presence of predicted 14-3-3 binding sites correlated with either an increase or decrease in microtubule binding under 14-3-3 inhibition (Fig 2B). If any changes under 14-3-3 inhibition are purely due to random statistical variations rather than 14-3-3 regulation, the number of proteins which increase or decrease in microtubule binding would be 50:50, regardless of whether proteins can bind 14-3-3. Among 1,504 proteins quantifiable in our experiments, 600 proteins had at least one site predicted at a high confidence to bind 14-3-3. Interestingly, among these 600 proteins, 375 increased (62%) and 225 decreased (38%) microtubule binding under 14-3-3 inhibition, showing proteins with predicted 14-3-3 binding sites are much more likely to increase microtubule binding (a difference of 150 proteins; $p < 0.001$; Fig 2B). In contrast, among the remaining 904 proteins without predicted 14-3-3 binding sites, 448 increased (49%) and 456 decreased (51%) microtubule binding, which is not significantly different from

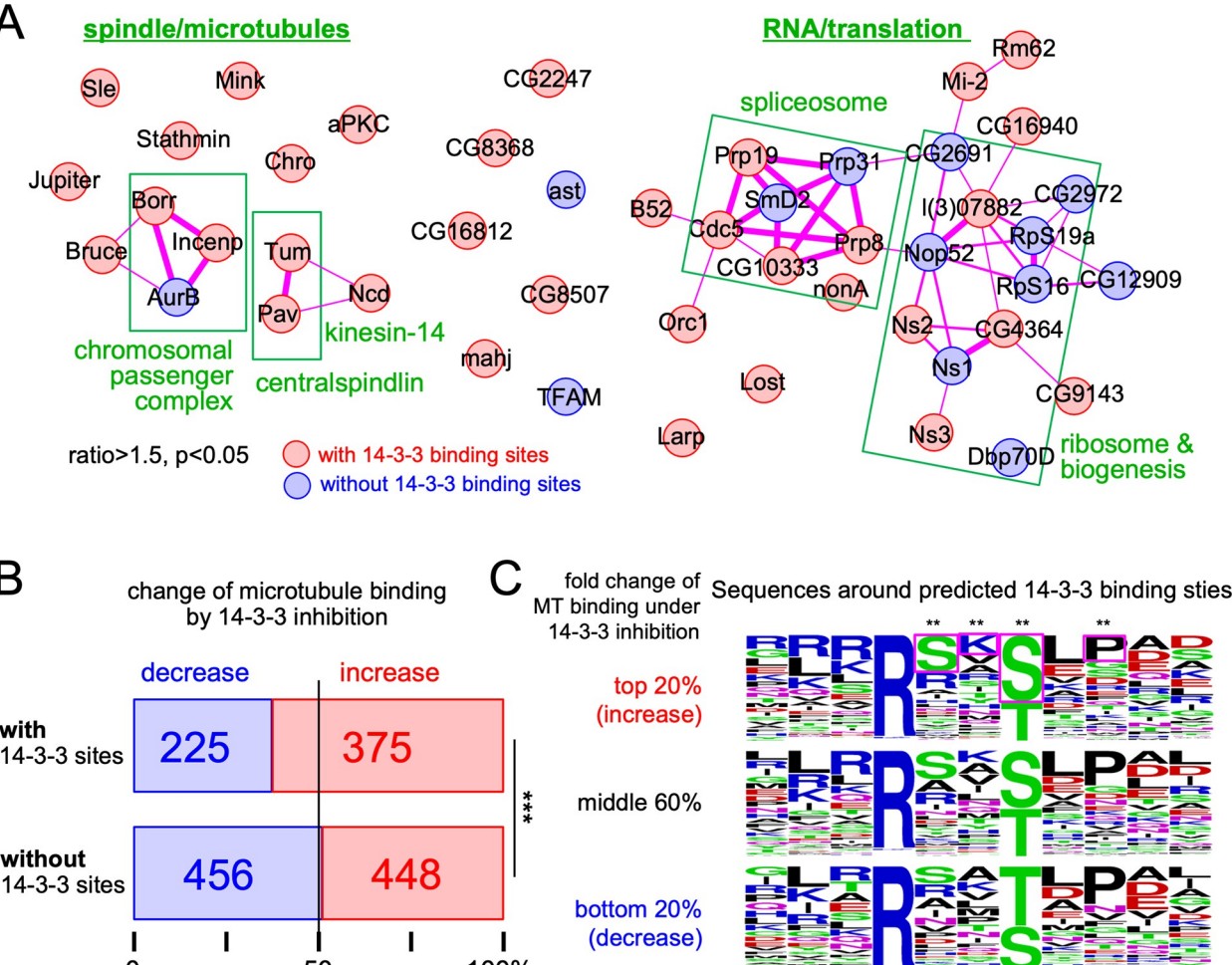

**Fig 2. Bioinformatics analysis.** (A) Physical protein-protein interactions known among 47 proteins that significantly increased their microtubule binding under 14-3-3 inhibition (the fold change >1.5 and p<0.05; the boxed area in Fig 1C). Red and blue indicate proteins with at least one predicted 14-3-3 binding sites and without them, respectively. The lines indicate physical interactions and their thicknesses indicate confidence levels of evidence. (B) The numbers and proportions of proteins with or without predicted 14-3-3 sites that increased or decreased in the microtubule fractions by 14-3-3 inhibition. (C) Sequences around predicted 14-3-3 binding sties among proteins with top 20%, middle 60% and bottom 20% of all detected proteins in the order of the fold change in microtubule fraction in the presence of the 14-3-3 inhibitor. Coloured boxes indicate the residues whose frequencies are significantly different among the top 20% than the frequencies among the rest of proteins. ** indicates p<0.01.

50:50 (p = 0.82; **Fig 2B**). This suggests that 14-3-3 potentially affects microtubule binding of many proteins in oocytes, and more often suppresses their microtubule binding than promotes it.

In the case of Kinesin-14/Ncd, 14-3-3 binding is regulated by phosphorylation at two sites, and regulating 14-3-3 binding is important for spatially controlling the microtubule binding activity of Kinesin-14/Ncd in oocytes. If 14-3-3 binding to a subset of proteins is regulated by a common mechanism such as phosphorylation by the same kinases, we may see a bias in sequences surrounding 14-3-3 binding sites in addition to the 14-3-3 binding motif. To gain an insight into regulation of 14-3-3 binding, sequences surrounding predicted 14-3-3 binding sites were compared between proteins whose microtubule binding is, or is not, regulated by 14-3-3. All identified proteins were divided into 5 bins according to the fold change in

microtubule association under 14-3-3 inhibition. Sequences around predicted 14-3-3 sites on proteins in each bin were pooled together. As the three middle bins are similar, the top 20% (increase under inhibition), the middle 60% and the bottom 20% (decrease under inhibition) were compared (**Fig 2C** and **S8 Table**). Serine at -2, serine at 0 and lysine at -1 are significantly overrepresented among the top 20%, while proline at +2 is underrepresented. As previously shown for Kinesin-14/Ncd, Serine at -2 can be phosphorylated by a kinase and this would negatively regulate 14-3-3 binding. Overrepresentation of serine over threonine at 0 (14-3-3 binding site) might also be interesting, as the phosphatase PP2A-B55 is known to prefer phospho-threonine over phospho-serine for dephosphorylation [29].

## Two phosphorylations regulate 14-3-3 interaction with the disordered region of Borealin

Our results showed that three out of the four CPC subunits (Aurora B, Incenp and Borealin) significantly increased their quantities in microtubule fractions under 14-3-3 inhibition (**Fig 1C**). Previous studies from us and others showed that the CPC has crucial roles in oocytes, including spindle microtubule assembly, spindle organisation, and bi-orientation of homologous centromeres [30–35]. The CPC dynamically localises to chromosomes, centromeres and the spindle equator in oocytes [33,36]. However, the CPC has not been reported to be regulated by 14-3-3 in any organism.

Among the CPC subunits, we identified 3 potential sites in Borealin and Incenp that match a consensus 14-3-3 binding sequence (**Fig 3A**) by bioinformatic analysis [37]. Among them, one site on Borealin (S163) not only matches the 14-3-3 binding consensus, but also shares a high similarity with the 14-3-3 binding sites of Kinesin-14/Ncd and Kinesin-6/MKlp1/Pav (**Fig 3B**). A distinct feature of these sites is the serine residue at the -2 position (S161) that matches the Aurora B phosphorylation consensus (R/KxS/T). Indeed, both S161 and S163 are phosphorylated *in vivo*, according to previous phosphoproteomics studies in *Drosophila* [38]. This potential 14-3-3 binding site (S163) on Borealin is located in a disordered region between the Incenp/Survivin interaction domain and the Borealin dimerisation domain [39,40] (**Fig 3A**).

To experimentally test whether 14-3-3 can bind Borealin at phosphorylated S163, this disordered region (residues 113–221) was produced with MBP-tag in bacteria (**S3 Fig**). As a control, the same region was produced with S163 mutated to a non-phosphorylatable alanine residue (S163A). After purification (**S3 Fig**), MBP-Borealin(113–221) and the S163A mutant were incubated with human PKD2 kinase. PKD2 was used as it is known to efficiently phosphorylate the similar 14-3-3 binding site in Kinesin-14/Ncd *in vitro* [18]. After the reaction, MBP-Borealin(113–221) and the S163A mutant were incubated with bacterially produced GST-14-3-3ε (**S3 Fig**). GST-14-3-3ε was then pulled down by glutathione-beads and analysed by western blot using an MBP antibody (**Fig 3C**). We found that MBP-Borealin(113–221) was efficiently pulled down with GST-14-3-3ε only after treatment with PKD2 (**Fig 3D**). In contrast, much less of the non-phosphorylatable S163A mutant was pulled down than wild-type Borealin even after PKD2 treatment (**Figs 3D and S4 and S9 Table**). This demonstrated that 14-3-3ε binds to Borealin at S163 in a phospho-dependent manner.

Next, we tested whether an additional phosphorylation by Aurora B can prevent Borealin from binding to 14-3-3, as a potential Aurora B phosphorylation site (S161) is located near the 14-3-3 binding site (S163). MBP-Borealin(113–221) was first incubated with PKD2, Aurora B, both kinases or no kinases, and then tested for 14-3-3 binding by pulling down with GST-14-3-3ε. We found that MBP-Borealin(113–221) phosphorylated by both PKD2 and Aurora B bound poorly to 14-3-3ε, much less than when phosphorylated by PKD2 alone

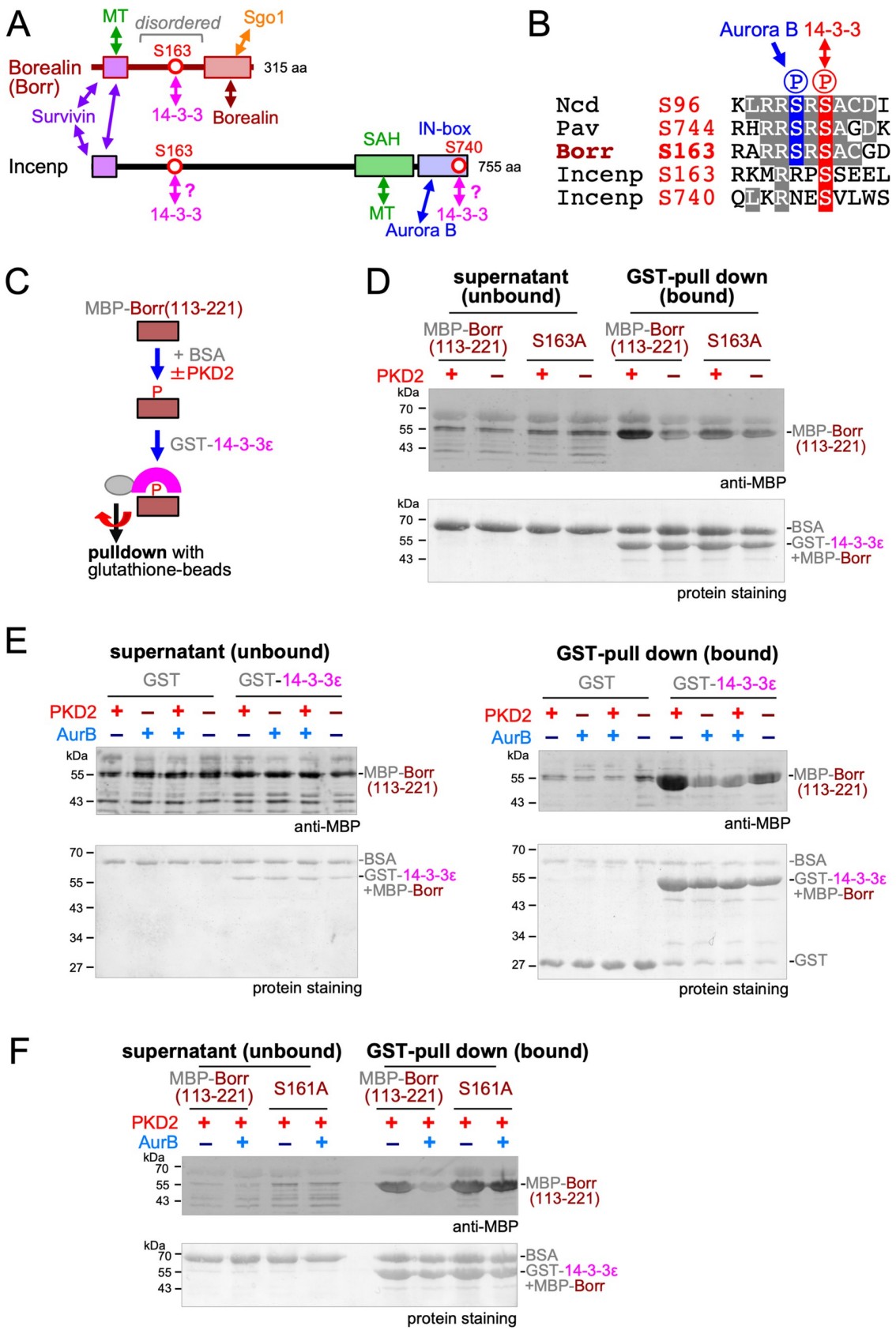

**Fig 3. 14-3-3 binding to the disordered region of Borealin is regulated by two phosphorylations.** (A) Diagram of domain organisation of Borealin and Incenp proteins. Known protein-protein interactions are indicated as double arrows. Three predicted 14-3-3 binding sites (red circles; Borealin S163 and Incenp S163 and S740) are identified among the CPC subunits. Borealin S163 is located within an intrinsically disordered region between domains interacting with the CPC subunits. MT; microtubule. (B) Sequences surrounding predicted 14-3-3 binding sites. The sequence surrounding Borealin has high similarity to Ncd S96 and Pav S744, containing serine at -2 position that is potentially phosphorylated by Aurora B kinase. (C) Testing interaction between the phospho-docking protein 14-3-3 and the disordered region of Borealin (Borr). Purified MBP-tagged Borealin fragment was incubated with or without PKD2, together with BSA. GST-14-3-3ε was added and pulled down by centrifugation using glutathione beads. Proteins bound to the beads and unbound fraction (supernatant) were analysed by western blotting using an MBP antibody. (D) MBP-Borealin(113–221) interacts with GST-14-3-3ε in a manner dependent on phosphorylation at S163. MBP-Borealin(113–221) and MBP-Borealin(113–221) with S163A mutation were incubated with or without human PKD2 kinase before pulldown by glutathione beads coupled with GST-14-3-3ε. A western blot of proteins bound to the glutathione beads and unbound fractions (supernatants) was carried out using an MBP antibody and total protein staining. Quantification and statistical analysis from triplicated experiments are shown in S4 Fig. (E) An additional phosphorylation by Aurora B prevents PKD2-phosphorylated MBP-Borealin(113–221) from interacting with GST-14-3-3ε. Borealin(113–221) was incubated with human PKD2 kinase alone, human Aurora B kinase alone, both kinases or without kinases, and tested for pull down using GST or GST-14-3-3ε. Bound or unbound proteins were analysed by a western blot using an MBP antibody or total protein staining. PKD2-phosphorylated MBP-Borealin(113–221) specifically interacted with GST-14-3-3ε, while MBP-Borealin(113–221) doubly phosphorylated by PKD2 and Aurora B did not. Quantification and statistical analysis from triplicated experiments are shown in S4 Fig. (F) Aurora B cannot prevent interaction between GST-14-3-3ε and PKD2-phosphorylated MBP-Borealin(113–221) with S161A mutation. MBP-Borealin (113–221) and MBP-Borealin(113–221) with S161A mutation were incubated with or without human Aurora B in the presence of human PKD2 kinase before pulldown by glutathione beads coupled with GST-14-3-3ε. A western blot of proteins bound to the glutathione beads and unbound fractions (supernatants) was carried out using an MBP antibody and total protein staining. Quantification and statistical analysis from triplicated experiments are shown in S4 Fig.

(**Figs 3E** and **S4** and **S9 Table**). This indicates that additional phosphorylation by Aurora B prevents 14-3-3ε interaction with Borealin already phosphorylated at S163. To confirm that this Aurora B phosphorylation is at S161, a non-phosphorylatable S161A mutation was introduced to MBP-Borealin(113–221). In a 14-3-3ε pulldown assay, Aurora B failed to prevent binding of 14-3-3ε to PKD2-phosphorylated MBP-Borealin(113–221) carrying this S161A mutation (**Figs 3F** and **S4** and **S9 Table**). Therefore, the additional Aurora B phosphorylation at S161 prevents binding of 14-3-3ε to Borealin phosphorylated at S163.

## Phospho-regulation of 14-3-3 binding is important for efficient Borealin localisation

Our biochemical analysis showed that 14-3-3 binding to Borealin is differentially regulated by two phosphorylations (**Fig 3B and 3E**). To define the *in vivo* role for this phospho-regulation in oocytes, we generated GFP-tagged versions of two non-phosphorylatable mutants at the 14-3-3 binding site (S163A) and the Aurora B phosphorylation site (S161A).

Their localisation was examined in the absence of the endogenous protein by expressing the GFP-tagged RNAi-resistant wild-type and non-phosphorylatable mutants together with a short hairpin RNA (shRNA) against the endogenous gene, and the chromosomal marker Rcc1-mCherry in oocytes. Live imaging showed that wild-type Borealin localised to the spindle equator and centromeres (**Fig 4A** and **S10 Table**), as reported previously for the CPC subunits [33,41].

A non-phosphorylatable Borealin mutant at the 14-3-3 binding site (S163A) nearly abolished the localisation on both the spindle equator and centromeres (**Fig 4A**). The non-phosphorylatable Borealin mutant at the Aurora B phosphorylation site (S161A) still localised, but with a reduced intensity, on both the spindle equator and centromeres in comparison to wild-type Borealin (**Fig 4A**). For quantification, a total intensity of GFP signals including both the spindle equator and centromeres above the background were measured. The S163A mutation resulted in a dramatic drop in the GFP signal intensity by 82% in comparison to wild-type Borealin (p<0.001), while the S161A mutation resulted in a less dramatic but significant reduction (by 59%; p<0.001) (**Fig 4B**). In contrast, intensities of the chromosome signal (Rcc1-mCherry)

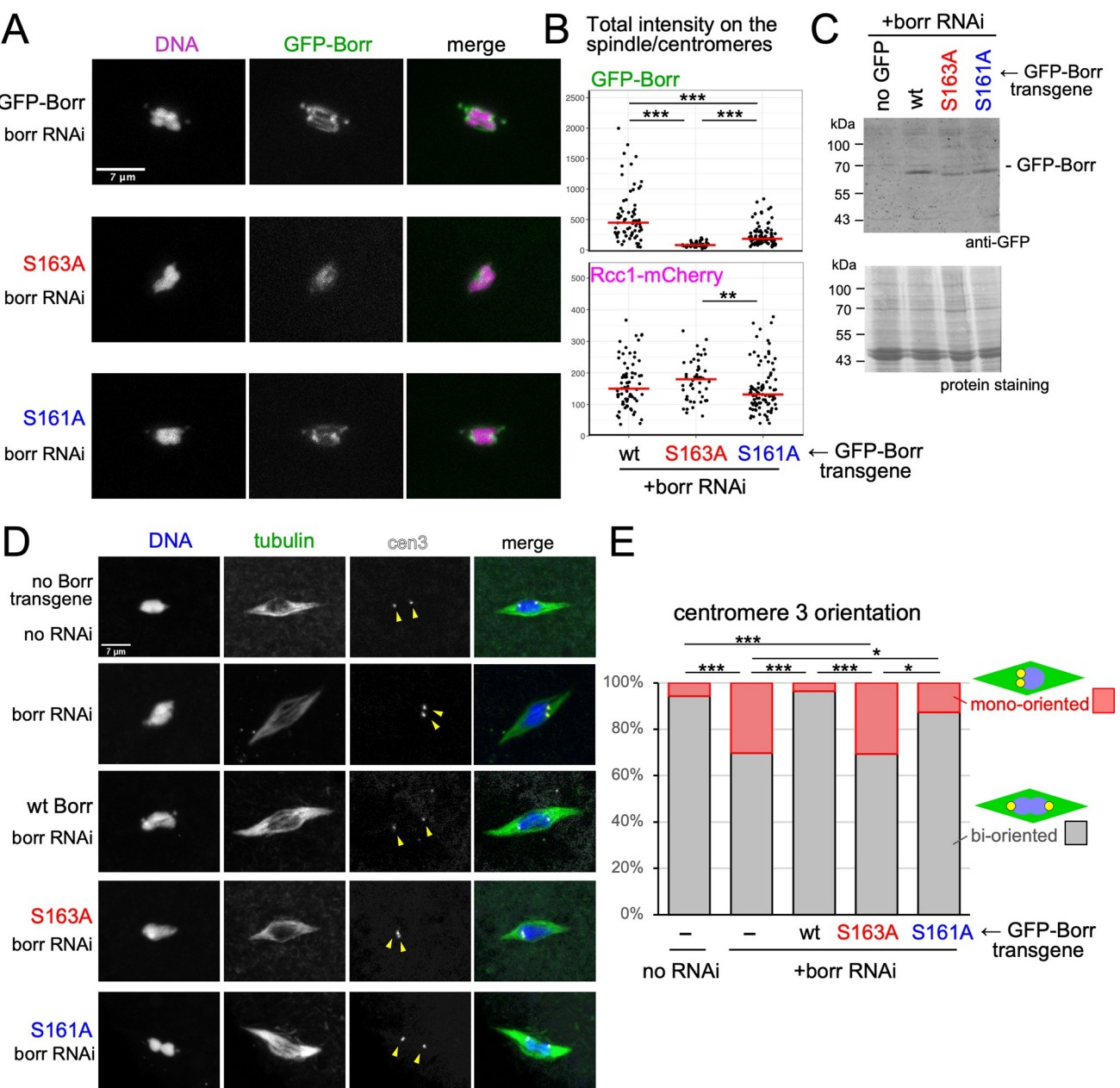

**Fig 4. Non-phosphorylatable mutations compromises the localisation of Borealin and bi-orientation of centromeres in oocytes.** (A) Non-phosphorylatable mutations (S163A, S161A) reduce the Borealin localisation to the spindle and centromeres. Fluorescence was observed in live oocytes expressing the GFP-tagged RNAi-resistant wild-type or non-phosphorylatable mutants together with a short hairpin RNA (shRNA) against the endogenous gene and a red chromosomal marker, Rcc1-mCherry. The images were presented using the same condition of capture and contrast adjustment for comparison. (B) The total signals of GFP-Borealin and Rcc1-mCherry on the spindle and centromeres above the background was quantified. Red lines indicate the median signal intensities. *** and ** indicate p<0.001 and p<0.01, respectively (Wilcoxon rank sum test). (C) Western blot of ovaries expressing GFP, wild-type ovaries without expression of GFP, and ovaries expressing wild-type or mutant GFP-Borr (S163A or S161A) in the background of *borr* RNAi, probed with an anti-GFP antibody and protein staining. (D) Non-phosphorylatable mutations (S163A, S161A) compromises the Borealin function in the bi-orientation of centromeres. α-tubulin and peri-centromere 3 (dodecasatellite; arrowheads) were visualised by Immunostaining combined with *in situ* hybridisation in *borr* RNAi oocytes with or without expression of the wild-type or non-phosphorylatable Borealin mutants. (E) Frequencies of bi-orientation of homologous centromeres of chromosome 3. Two separate dodecasatelite signals near both ends of the chromosome mass are considered as bi-oriented centromeres, and one or closely located two signals on the one side of the chromosome mass are considered as mono-oriented centromeres. *** and * indicate p<0.001 and p<0.05, respectively (Fisher exact test).

in the phospho-mutants were not significantly different from those in wild-type Borealin, suggesting that the difference in Borealin signal intensity is due to the reduction in localisation, rather than differences in imaging (**Fig 4B**). Immunoblotting suggested that these differences in signal intensity are not explained by the expression levels of GFP-Borealin mutant proteins (**Fig 4C**).

These results revealed that phospho-regulation of 14-3-3 binding to Borealin is important for Borealin localisation in oocytes. The phospho-dependent 14-3-3 binding is crucial for Borealin localisation, while the phosphorylation by Aurora B is less crucial but still important.

## 14-3-3 binding to Borealin is important for bi-orienting homologous centromeres

To establish the functional significance of the phospho-regulation of 14-3-3 binding in oocytes, we further examined the oocytes expressing the wild-type or mutant Borealin transgene in a background in which the endogenous gene is silenced. We then examined spindle morphology and centromere positions in mostly metaphase I arrested oocytes by immunostaining tubulin combined with *in situ* hybridisation using a peri-centromere probe specific to chromosome 3 (dodecasatellite; [42]).

We examined oocytes expressing an shRNA against Borealin and without Borealin transgenes alongside a control without shRNA expression (**Fig 4D and 4E**). In the control without shRNA expression, one centromere 3 signal was found on each side of the chromosome mass in nearly all (95%) oocytes, showing a pair of homologous centromeres that are bi-oriented and pulled apart to both poles. Only 5% of oocytes showed a mono-orientation in the control. In contrast, 30% of RNAi oocytes showed mono-orientation (p<0.001). The spindle morphologies look similar between Borealin RNAi oocytes and the control. This is likely to reflect partial depletion of Borealin, although we cannot exclude the possibility that Borealin is not essential for normal spindle morphology.

Expression of GFP-tagged wild-type Borealin fully rescued the centromere bi-orientation defect of Borealin RNAi, and the frequency of mono-orientation went back to a similar level to oocytes without RNAi. In contrast, expression of GFP-tagged Borealin(S163A) failed to rescue the defect, and the frequency of mono-orientation stayed at a similar level to the RNAi oocytes without a transgene. This is consistent with the large reduction in localisation to the spindle/centromeres we observed above (**Fig 4D and 4E** and **S11 Table**). Expression of GFP-tagged Borealin(S161A) showed an intermediate effect, which is also consistent with partial loss of localisation. These results showed that phosphorylation for 14-3-3 binding at S163 is essential for Borealin function, while phosphorylation of the flanking site (S161) is not essential but appears to play a role. These results showed that phospho-regulation of 14-3-3 binding is important for the centromere bi-orientation function of Borealin.

## Discussion

By microtubule co-sedimentation combined with quantitative mass-spectrometry, we identified various proteins whose microtubule binding is regulated by 14-3-3 in *Drosophila* oocytes. Among them, we showed that 14-3-3 binds to Borealin, a subunit of the chromosomal passenger complex (CPC). Two phosphorylations regulating this 14-3-3 binding to Borealin are important for Borealin localisation and centromere bi-orientation in oocytes.

Using a newly developed method, we quantified the relative amounts of over 1,500 proteins associated with microtubules in ovary extract with or without 14-3-3 inhibition. Our analysis showed that 14-3-3 suppresses microtubule binding of a substantial number of proteins in ovaries. Among these proteins, Kinesin-14/Ncd and the centralspindlin subunits were previously

reported to be regulated by 14-3-3 [18,28] in *Drosophila* oocytes and human cells, respectively. We also identified three subunits of the CPC, one of the master regulators of cell division [32,43]. To our knowledge, this is the first report to show that the CPC is regulated by 14-3-3.

We found that the CPC subunit Borealin has a predicted 14-3-3 binding site (S163) in its disordered middle region. The surrounding sequence shares a striking similarity to the 14-3-3 binding sites identified in Kinesin-14/Ncd and the centralspindlin subunit MKlp1/Pavarotti [18,28]. Through *in vitro* studies, we showed that 14-3-3ε binds to Borealin at S163 in a phospho-dependent manner, although the kinase that phosphorylates this site in oocytes is unknown. Additional phosphorylation by Aurora B, probably at S161, prevents 14-3-3ε binding, in a similar manner to that reported for Kinesin-14/Ncd [18]. A non-phosphorylatable mutation predicted to abolish 14-3-3 binding (S163A) dramatically reduced the amount of Borealin at both the spindle and centromeres. This mutation also abolishes the function of Borealin in bi-orienting homologous centromeres. On the other hand, a non-phosphorylatable mutation of the predicted Aurora B site (S161A) also compromises the normal localisation and function of Borealin but does not abolish either. This partial requirement of the Aurora B phosphorylation can be explained if 14-3-3 binding to Borealin is prevented by redundant action of two enzymes: a phosphatase dephosphorylating the 14-3-3 binding site (S163) and Aurora B phosphorylating the neighbouring site (S161). Similar effects were also observed in the equivalent mutations in Kinesin-14/Ncd [18], strengthening our hypothesis that a common mechanism regulates Borealin and Kinesin-14/Ncd.

Based on the model proposed by Beaven et al [18] for Kinesin-14/Ncd, we propose that Borealin is prevented from binding microtubules in the oocyte cytoplasm by its association with 14-3-3. When it is in the vicinity of the chromatin, Aurora B kinase activity reverses the 14-3-3 binding and allows the CPC to bind spindle microtubules. We showed that this regulation is important for CPC's role in bi-orienting homologous centromeres. In our model, spatial regulation mediated by 14-3-3 and Aurora B allows Borealin to bind selectively to spindle microtubules. Abolishing this spatial regulation would dramatically reduce the effective concentration of Borealin near the spindle by binding to numerous non-spindle microtubules in the large volume of the oocyte. This would result in a reduction of Borealin localisation to both centromeres and spindle microtubules. However, we cannot rule out that 14-3-3 and Aurora B may regulate Borealin localisation to both centromeres and spindle microtubules differently from Kinesin-14/Ncd.

This study has established 14-3-3 as a general regulator of microtubule-associated proteins (MAPs). 14-3-3 may play an important role in spatially regulating many MAPs in oocytes in combination with other regulators such as Aurora B or phosphatases, in parallel to the Ran/importin system. Future studies of these 14-3-3 regulated proteins will shed light on how microtubules are regulated in oocytes. The study has also revealed a novel regulation of the CPC. The CPC is one of the conserved master regulators of cell division, and dynamically localises to chromosomes, centromeres and spindle microtubules. Although the chromosome/centromere localisation of the CPC has been intensely studied, microtubule binding and its regulation are paid much less attention. Our study identified a novel mechanism in which 14-3-3 regulates the microtubule binding activity of the CPC. However, the CPC is known to have other microtubule binding domains than this disordered region of Borealin [44] and Incenp may also bind 14-3-3 at the sites we show above. Further studies are required to establish whether this region of Borealin contains a new microtubule binding domain and how multiple microtubule binding sites and 14-3-3 binding sites may work together to regulate the localisation and microtubule binding of the CPC as a whole.

Beyond meiosis, the same mechanism may also operate during cytokinesis when the CPC accumulates to the midbody. As our proposed mechanism can provide a self amplification

loop by which the CPC promotes further recruitment of the CPC, in theory this could confer a localised switch-like property to CPC activity during cytokinesis and/or help to accumulate the CPC to a tighter region.

## Materials and methods

### Fly maturation and generation of GFP-Borr flies

Standard fly techniques were followed [45]. *Drosophila melanogaster* stocks were cultured on standard cornmeal medium at 25°C. $w^{1118}$ was used as wild-type. Less than 1 day old female adults were matured for 3 days at 25°C in the presence of males before dissection of ovaries.

To generate Borealin transgenes, the full-length Borr open reading frame was amplified from LD36125 by PCR using PrimeStar (Takara) and cloned between AscI and NotI sites of pENTR (ThermoFisher) using Gibson assembly (HiFi; NEB). To make the gene resistant to RNAi, silent mutations (from GCG GTG TTC to GCC GTC TTT) were introduced by PCR using primers containing the mutations and followed by Gibson assembly. Further mutations (TCC to GCC for S163A; AGT to GCC for S161A) were introduced using the same method. The absence of unwanted mutations were confirmed by Sanger sequencing. These entry clones were recombined with the destination vector φMGW [36] by LR Gateway Clonase II (ThermoFisher).

Transgenic flies expressing RNAi-resistant GFP-Borr (wild-type/S163A/S161A) under the maternal α -tubulin promoter were generated using φC31 integrase-mediated transgenesis at the *VK18* site [46] on the second chromosome, performed by BestGene Inc. Each transgene of GFP-Borr or variant was then recombined with another transgene at the *attP40* site [47] expressing short hairpin (sh) RNA against *borr* (HMC04381; [48]; 55942 from Bloomington Drosophila Stock Center).

### Microtubule cosedimentation with the 14-3-3 inhibitor R18

Microtubules and their associated MAPs were purified from ovaries by modifying protocols previously used in embryos [49] and in S2 cells [23]. Ovaries from matured female $w^{1118}$ flies were dissected in 1x BRB80 buffer (80 mM PIPES-KOH pH 6.8, 1 mM MgCl$_2$, 1 mM Na$_3$EGTA) supplemented with phosphatase inhibitors (1mM DTT (Promega), 1mM PMSF, Complete EDTA-free Protease Inhibitor Mixture Tablets MINI, diluted according to manufacturer instructions (Roche), 15 mM Na$_3$VO$_4$, 10 mM p-nitrophenyl phosphate, 1 μM okadaic acid). Ovaries were snap frozen in liquid nitrogen and stored at -80°C. About 100 pairs of ovaries (~150 mg in wet weight) were defrosted and pooled. An equal volume of 2x BRB80+-phosphatase inhibitors was added. Ovaries were homogenised in a pre-cooled Dounce homogeniser and incubated on ice for 30 minutes to depolymerise microtubules, then cleared by centrifugation at 13,000 rpm/13,628 x g for 15 minutes at 4°C. The supernatant was cleared again by 2 rounds of ultracentrifugation at 100,000 rpm (358,400 x g; Beckman TLA-120.2) for 10 minutes at 0°C, and then incubated at 30°C for 20 minutes, followed by ultracentrifugation at 100,000 rpm for 10 minutes at 22°C. Actin was depolymerised by the addition of 1 μg/mL Latrunculin A (Cambridge Bioscience), 2 μg/mL Latrunculin B (Cambridge Bioscience), 0.1 mM DTT (Promega) and 1 mM GTP (Sigma), and an aliquot of the lysate was kept for analysis ('input').

The remaining supernatant was split into two portions, one of which ('sample') was treated with R18 peptide (R18 trifluoroacetate, Sigma) at a final concentration of 100 μM to inhibit 14-3-3 activity. To the other ('control') was added an equal volume of water. Both were then incubated for 5 minutes at room temperature. Paclitaxel (Sigma-Aldrich) was subsequently added to both sample and control at a final concentration of 20 μM, and incubated for 30

minutes to allow microtubules to polymerise. Microtubules were pelleted twice through 50% sucrose cushions containing 20 μM paclitaxel via ultracentrifugation at 50,000 x g for 35 minutes at 22˚C. Pellets were resuspended in 50 μl of 1x BRB80 + phosphatase inhibitors supplemented with 20 μM paclitaxel. An equal volume of 3x SDS sample buffer + 5% β-mercaptoethanol was added to samples before boiling at 95˚C for 2 minutes to denature proteins. Samples were stored at -20˚C. Aliquots of samples were analysed by western blot. Total proteins on the membrane were stained with MemCode reversible staining kit (Thermo-Fisher) first, and then with a rat polyclonal antibody against α-tubulin (1:500) followed by IRDye 800CW conjugated goat anti-mouse IgG antibody (LI-COR). The signals were detected with an Odyssey CLx imaging scanner (LI-COR). The brightness and contrast were adjusted uniformly across the entire area in a linear manner without removing or altering features.

## Mass spectrometry

Pellets from microtubule cosedimentation were separated using a NuPAGE 12% Bis-Tris gel with MOPS running buffer and stained with Colloidal Blue Staining Kit (Invitrogen). 2 lanes were run for each pellet. The gel region containing the tubulin band was excised and discarded from both sample and control lanes. The remaining gel regions were trypsin-digested and reduced/alkylated using standard procedures [50]. Following digestion, samples were diluted with an equal volume of 0.1% TFA and spun onto StageTips as described by Rappsilber et al. [51]. Peptides were eluted in 80% acetonitrile in 0.1% TFA and concentrated 40x by vacuum centrifugation.

Samples were prepared for LC-MS/MS analysis by diluting them to 5 μL with 0.1% TFA. MS-analysis was performed on an Orbitrap Fusion Lumos tribrid mass spectrometer (Thermo Fisher Scientific, UK) coupled on-line to Ultimate 3000 RSLCnano Systems (Dionex, Thermo Fisher Scientific, UK). Peptides were separated by a PepMap RSLC C18 EasySpray column (2 μm, 100Å, 75 μm x 50 cm) (Thermo Fisher Scientific, UK), operating at 50˚C. Parameters are described in **S12 Table**. The peptide gradient was: 2 to 40% buffer B in 140 min, then to 95% in 11 min. The percentage of buffer B remained at 95 for 5 minutes and returned back at 2 one minute after. Peptides were selected and fragmented by higher-collisional energy dissociation (HCD) [52] with normalised collision energy of 30. Raw files were processed using MaxQuant software platform [53] version 1.5.2.8 against the complete *Drosophila melanogaster* proteome (Uniprot, released in September 2016), using Andromeda [54]. Raw mass spectrometry proteomics data were deposited to the ProteomeXchange Consortium (http://proteomecentral.proteomexchange.org) via the PRIDE partner repository [55] with the dataset identifier PXD030445.

## Bioinformatic analysis

Peptides were assigned to Uniprot accession numbers, and arranged as one accession per row. When more than one accession was assigned per gene, the accession with the lower score was removed. Accessions assigned only one peptide cannot be confidently identified and so were removed. Accessions denoting different RNA transcripts from the same gene were collapsed into one accession, again keeping the highest scored accession in the event of duplicates. The ratio between sample and control intensities for each gene was calculated as $\log_2(\text{sample})$–$\log_2(\text{control})$. When either the sample or control intensity was 0, these accessions were removed from the graph and the statistical analysis and analysed separately.

To determine the consistency between replicates, $\log_2(\text{sample})$ and $\log_2(\text{control})$ value distributions were compared between replicates, as well as the line of best fit for the graph of $\log_2(\text{sample})$ vs $\log_2(\text{control})$. 6 out of 8 replicates were determined to be similar enough to

continue with analysis. To compare the sample and control intensities for each accession, the given accession had to be detected in at least two replicates for both sample and control. The ratios were calculated as above, and the p-value for each ratio calculated using a two-tailed unpaired t-test. Then the ratio and p-value for each protein were plotted in $\log_2$ as a volcano plot (**Fig 2C**).

Gene ontology terms were downloaded from Flybase [56]. Data manipulation and statistical analyses were performed using Excel (Microsoft) and RStudio (RStudio Team, 2020) using the R scripting language [57]. Graphs were plotted using ggplot2 [58]. Candidate 14-3-3 binding sites were predicted using the 14-3-3 Pred tool [37] on the basis of a consensus score above 0.9 and individual scores above the default thresholds (ANN 0.55; PSSM 0.80; SVM 0.25) for all 3 prediction methods. Likely kinase target sequences were predicted in Borealin sequence using the GPS 3.0 programme [59] with medium threshold. Protein fasta sequences were obtained from UniProt. Protein interaction data for network analysis was downloaded from STRING-db [27] on 29 Nov 2021. Included interactions were limited to physical subnetwork above medium confidence (0.400) sourced from experiments and databases. 14-3-3 site sequence comparison analysed and presented using Weblogo (https://weblogo.berkeley.edu/, [60]).

## GST-14-3-3 binding to MBP-Borr including phosphorylation

To generate a construct expressing MBP-Borr(113–221) in *E. coli*, Borr open reading frame corresponding to amino acids 113–221 was first amplified with a stop codon from LD36125 by PCR using PrimeStar (Takara) and cloned between AscI and NotI sites of pENTR (Thermo-Fisher) using Gibson assembly (HiFi; NEB). The S161A mutation (AGT to GCC) or the S163A mutation (TCC to GCC) was introduced to this entry clone by PCR-amplifying the entire plasmid using primers carrying the mutation followed by Gibson assembly (HiFi; NEB). These entry clones were recombined using LR clonase II (ThermoFisher) with a destination vector modified from pMAL-c2 (NEB) by insertion of the Gateway cassette (ThermoFisher) at the XmnI site. GST-14-3-3ε was produced using a construct previously reported [18].

To test if Borealin is bound by 14-3-3, MBP-Borr(113–221) and GST-14-3-3ε were purified from *E. coli* (BL21/pLysS) carrying the expression constructs using amylose beads (NEB) and S-glutathione beads (GE Healthcare), respectively. Fractions were eluted using 10 mM reduced glutathione through a polypropylene column (Qiagen), and those with the highest concentrations combined. Combined elutes were then dialysed overnight using Slide-a-Lyser dialysis cassettes (Thermo) in 1 l of dialysis buffer (50 mM Na-phosphate buffer pH7.6, 250 mM KCl, 1 mM $MgCl_2$, 5 mM β-mercaptoethanol). Subsequently, the dialysed purified GST-14-3-3ε was concentrated to 500 µl total using Ultra-4 Spin Columns (Amicon), before being aliquoted, supplemented with 10% final concentration of glycerol, snap frozen in liquid nitrogen and stored at -70˚C.

After thawing on ice, 3 µg (3.75 µl) of purified MBP-Borr(113–221) protein (wild-type, S161A and S163A) were cleared of aggregates by centrifugation for 1 minute at 14,000 rpm (13,628 x g), and incubated with 20 µl washed S-glutathione beads for 30 minutes at 4˚C. After the beads were removed by centrifugation, 7.5 µg of MBP-Borr(113–221) was incubated with either 450 ng Aurora B (Cambridge Biosciences), 337.5 ng of PKD2 (Cambridge Biosciences), both Aurora B and PKD2 or no kinase. Phosphorylation was carried out in a two-step incubation at 30˚C in phosphorylation buffer (20 mM HEPES pH 7.4, 2 mM $MgCl_2$, 1 mM ATP, 27.5 mM KCl, 1 mM DTT, 0.2 mg/mL BSA, 1 mM EGTA), with Aurora B added for the first 30 minutes, followed by addition of PKD2 and a further 60 minutes at 30˚C. Phosphorylated fragments (5 µl) were then mixed with 20 µg GST or GST-14-3-3 in 500 µl Pulldown buffer (25 mM Tris-Cl pH = 7.6, 150 mM NaCl, 0.5% Triton X-100, 0.3 mM $Na_3VO_4$) and allowed to

bind for 30 minutes on ice. They were then incubated with 20 μl washed S-glutathione beads on a rotator at 4˚C for 1 hour. The supernatant/unbound fraction was removed and kept for analysis, and beads were then washed 3 times in 800 μl of Pulldown buffer. 2x SDS loading buffer was added to beads and supernatant samples at 1:1 ratio, supplemented with 5% final volume of β-mercaptoethanol, and were boiled for 2 minutes at 95˚C, before analysis by western blot. Proteins were separated by SDS-PAGE and transferred onto a nitrocellulose membrane using the Mini-PROTEAN and Mini Trans-Blot system (Biorad). Total proteins on the membrane were stained with MemCode reversible staining kit (Thermo-Fisher). After being destained, the membrane was incubated with a rat polyclonal antibody against MBP (1:500) followed by IRDye 800CW conjugated goat anti-rat IgG antibody (LI-COR). The signals were detected with an Odyssey CLx imaging scanner (LI-COR). The brightness and contrast were adjusted uniformly across the entire area in a linear manner without removing or altering features. The band intensities over the background were quantified by the following method. Two rectangular boxes (S and L) were drawn. Box S includes mainly the band of interest, and Box L includes the band and background areas above and below the band. The total signal intensity over the background was calculated by the formula $I_S - (N_S \times [I_L - I_S] / [N_L - N_S])$, where I and N are the total pixel intensity and pixel number in the specified box (S or L), respectively.

## Live-imaging of oocytes

To visualise the localisation of GFP-Borr in live oocytes, flies carrying two transgenes that express GFP-tagged Borr (wild-type, S163A or S161A) under the maternal α-tubulin promoter and shRNA against *borr* under the *UASp* promoter were crossed with flies carrying *GAL4-VP16* (*V37*; Bloomington Drosophila Stock Centre 7063) and *Rcc1-mCherry* [36] both under the maternal α-tubulin promoter. Ovaries from 3-day matured flies were dissected one at a time in Halocarbon oil (700; Halocarbon) on a cover slip. Stage 14 oocytes were identified by their morphology, namely a well-formed chorion and long dorsal appendages. Oocytes which had entered the oviduct were discarded along with other stages. These oocytes were observed under a microscope (Axiovert 200M; Zeiss) attached to a spinning disk confocal head (CSU-X1; Yokogawa) controlled by Volocity (PerkinElmer). A Plan-Apochromat objective lens (63x/1.4 numerical aperture) was used with Immersol 518F oil (Zeiss). Images were taken with a Z-slice interval of 0.8 μm and using 40% laser intensity in the red channel, 100% intensity in the green channel. Maximum intensity projections of the Z-stacks are presented as figures and were used for analyses. Images were exported in the TIFF format and signal intensity measurements carried out in ImageJ as followed. The total signal intensities of GFP-Borealin and Rcc1-mChery on the spindle were estimated using the following method. Two areas (S and L) were drawn on the maximum-intensity projection made from Z series of images. Area S includes mainly the central spindle and centromeres or all chromosomes, and Area L includes this area and surrounding region. The total signal intensity over the background was calculated by the formula $I_S - (N_S \times [I_L - I_S] / [N_L - N_S])$, where I and N are the total pixel intensity and pixel number in the specified area (S or L), respectively.

## Immunostaining and fluorescence *in situ* hybridisation

For immunostaining, ovaries were dissected out of mature flies in methanol as previously described [61]. Sonication was used to disrupt remove the chorion and, which can interfere with antibody penetration. Ovaries were sonicated at 38% amplitude (Vibra Cell VCX500; Sonics) for three 1-second pulses and selected for removal of chorion and vitelline membranes. This was repeated by a further one or two times. The resultant oocytes were washed in 1x PBS,

and blocked for 1 hour in blocking buffer (10% foetal calf serum in PBS-T) on a rotator. Oocytes were incubated overnight with primary antibodies diluted in blocking buffer. After washing with PBS-T, fluorophore-conjugated secondary antibodies (Alexa 488, 1:250, or Cy3, 1:1000, in PBS-T) along with DAPI (0.2 μg/ml, Sigma) were added, and incubated for 2 hours in the dark. Finally, oocytes were washed in PBS-T and mounted on microscopy slides in glycerol.

For FISH combined with immunostaining in oocytes, stage 14 oocytes were prepared as for immunostaining, and then post-fixed in 8% formaldehyde as described in [62,63]. An oligonucleotide (CCCGTACTGGT)$_4$ for dodecasatellite [42] near centromere 3 was used. 100 pmol of probe was end-labelled with 2 nmol Alexa546-conjugated dUTP (Invitrogen), 16 nmol of unlabelled dTTP (Promega) and 30 units of terminal deoxynucleotidyl transferase (Promega) in 20 μl of transferase buffer at 37˚C for one hour. The reaction was halted by incubation at 70˚C for 10 minutes, and then remaining free dTTP was removed using a G25 Mini Quick spin column (Roche). 4 μl of this labelled oligonucleotide was added to ovaries in 40 μl of hybridisation buffer (0.1 g/ml dextran sulphate, 50% formamide, 3xSSC) at 30˚C overnight. After washing twice in washing buffer (50% formamide, 2x SSC, 0.1% Triton X100) at 30˚C for total 30 minutes, oocytes were further washed three times in 2xSSC+0.01% Triton X100 and once in PBS. After blocking in PBS+0.1% Triton containing 10% fetal calf serum, the immunostaining procedure was followed for antibody staining.

These slides were then observed via a confocal microscopy (LSM800 on AxioObserver Z1; Zeiss) using a Plan-Apochromat objective lens (63x/1.4 numerical aperture) with Immersol 518F oil (Zeiss). Images spanning an entire spindle were taken with a Z-slice interval of 0.5 μm and Zoom 2.0 (pixel size 0.1 μm). Maximum intensity projections of the Z-stacks are presented as figures.

Expression of GFP-Borr in ovaries were assessed by western blot using a rabbit anti-GFP antibody (1:100, Invitrogen, A11122) and IRDye 800CW conjugated goat anti-mouse IgG antibody (LI-COR). The signals were detected with an Odyssey CLx imaging scanner (LI-COR). MemCode reversible staining kit (Thermo-Fisher) was used to visualise any proteins. The brightness and contrast were adjusted uniformly across the entire area in a linear manner without removing or altering features.

## Supporting information

**S1 Fig. Microtubule co-sedimentation from *Drosophila* ovaries.** (A) Microtubules and associated proteins were co-sedimented with or without the 14-3-3 inhibitor R18 from soluble extract of *Drosophila* ovaries. The original supernatant, wash of the original pellet, and the final pellet used for mass-spectrometry were analysed by western blot using an α-tubulin antibody and total protein staining. (B) The final pellets were run on SDS-PAGE and stained with Coomassie for mass-spectrometry. Nearly all tubulin in the extract was found in the pellet fraction, which predominantly consists of tubulin with minimal amounts of other proteins, regardless of the presence or absence of R18.
(PDF)

**S2 Fig. Physical protein-protein interactions known among 20 proteins with significantly lower amounts in the microtubule fraction under 14-3-3 inhibition.** Proteins with the fold change <0.6667 and the p<0.05 are shown. Red and blue indicate proteins with at least one predicted 14-3-3 binding site and without any, respectively. Lines indicate known physical interactions, and they do not have significantly more interactions than expected.
(PDF)

**S3 Fig. Protein purification of GST, GST-14-3-3ε, MBP-Borr(113–221) and MBP-Borr (113–221,S163A) from *E. coli*.** GST (A), GST-14-3-3ε (B) MBP-Borr(113–221) (C), MBP-Borr(113–221,S163A) (D) and MBP-Borr(113–221,S161A) (E) were affinity-purified using columns containing S-glutathione beads or amylose beads and eluted in buffer containing glutathione or maltose, respectively, as described in Materials and Methods. Purification intermediates were analysed by SDS-PAGE and stained with Coomassie.
(PDF)

**S4 Fig. Quantification of MBP-Borr in the GST-14-3-3ε pull down in triplicated experiments.** Experiments shown in Fig 3D–3F were triplicated and presented as different coloured bars. The total signal intensities of the MBP-Borealin bands above the background were normalised to that of PKD2-phosphorylated MBP-Borr in GST-14-3-3 pull down in the same experiment. **, * and *ns* indicate $p<0.01$, $p<0.05$ and $p>0.05$, respectively. (A) MBP-Borealin (113–221) interacts with GST-14-3-3ε in a manner dependent on phosphorylation at S163. (B) An additional phosphorylation by Aurora B prevents PKD2-phosphorylated MBP-Borealin (113–221) from interacting with GST-14-3-3ε. (C) Aurora B cannot prevent interaction between GST-14-3-3ε and PKD2-phosphorylated MBP-Borealin(113–221) with S161A mutation.
(PDF)

**S1 Table. Mass spectrometry intensity data for all proteins in microtubule fractions with and without R18.** This consists of 3 sheets containing all quantifiable proteins, 47 proteins that significantly increased their microtubule binding under 14-3-3 inhibition (the fold change $>1.5$ and the $p<0.05$) and 20 proteins that significantly decreased their microtubule binding under 14-3-3 inhibition (the fold change $<0.6667$ and the $p<0.05$). **geneName:** Name/symbol of gene as listed on Flybase. **Accession.No:** Uniprot accession number, as assigned by Max-Quant. **logSAM:** The average of all Sample (R18-treated) $\log_2$ intensity values. **logCON:** The average of all Control (untreated) $\log_2$ intensity values. **pairedp.value :** The p-value from a paired Student's T-test. **logSAM1:** $\log_2$ intensity value from Sample (R18-treated) replicate 1. **logCON1:** $\log_2$ intensity value from Control (untreated) replicate 1. **logSAM2:** $\log_2$ intensity value from Sample (R18-treated) replicate 2. **logCON2:** $\log_2$ intensity value from Control (untreated) replicate 2. **logSAM5:** $\log_2$ intensity value from Sample (R18-treated) replicate 5. **logCON5:** $\log_2$ intensity value from Control (untreated) replicate 5. **logSAM6:** $\log_2$ intensity value from Sample (R18-treated) replicate 6. **logCON6:** $\log_2$ intensity value from Control (untreated) replicate 6. **logSAM7:** $\log_2$ intensity value from Sample (R18-treated) replicate 7. **logCON7:** $\log_2$ intensity value from Control (untreated) replicate 7. **logSAM8:** $\log_2$ intensity value from Sample (R18-treated) replicate 8. **logCON8:** $\log_2$ intensity value from Control (untreated) replicate 8. **logratio: (**the average of $\log_2$SAM)–(the average of $\log_2$CON)
(XLSX)

**S2 Table. Raw mass spectrometry intensity data for replicate 1.** Sheet 1 ("proteinGroups") lists all protein groups identified in the experiment. Sheet 2 ("CON, REV filtered") lists groups after filtering (using reverse decoy database, etc). Sheet 3 ("LFQ") is a summary using data from sheet 2. **Sheet 1: "proteinGroups" and sheet 2: "CON, REV filtered"**. **Protein IDs**: Ensembl or Uniprot Identifiers of a group of proteins with overlapping sequence, i.e. cannot be distinguished from one another with peptides from this experiment. **Majority protein IDs**: Proteins with at least half the peptide matches of the leading protein in the group. **Peptide counts (all)**: The number of peptides assigned to this protein group. **Peptide counts (razor+-unique)**: The number of unique peptides (not shared with other groups) + razor peptides (can be shared with other groups) assigned to each protein in this group. **Peptide counts (unique)**:

The number of unique peptides (peptides which only match this group and no other). **Fasta headers**: The Uniprot fasta headers for the members of this group, containing accession number, name, description and mass. **Number of proteins**: The number of proteins in this group. **Peptides**: The total number of peptides assigned to this group. **Razor + unique peptides**: The number of razor + unique peptides assigned to this group. **Unique peptides**: The number of unique peptides (only assigned to this group). **Peptides CON1 (or CON2. . . etc)**: The number of peptides assigned to this group, in the control (untreated). **Peptides SAM1 (or SAM2. . . etc)**: The number of peptides assigned to this group, in the sample (R18-treated). **Razor + unique peptides CON1**: The number of razor + unique peptides assigned to this group in the control (untreated). **Razor + unique peptides SAM1**: The number of razor + unique peptides assigned to this group in the sample (R18-treated). **Unique peptides CON1**: The number of unique peptides assigned to this group in the control (untreated). **Unique peptides SAM1**: The number of unique peptides assigned to this group in the sample (R18-treated). **Sequence coverage [%]**: Percentage of the lead protein sequence covered by peptides. **Unique + razor sequence coverage [%]**: Percentage of the lead protein sequence covered by unique + razor peptides. **Unique sequence coverage [%]**: Percentage of the lead protein sequence covered by unique peptides only. **Mol. weight [kDa]**: Molecular weight of the protein in kDa. **Sequence length**: Length of the lead protein sequence. **Sequence lengths**: Lengths of all proteins in the group. **Q-value**: The ratio of forward to reverse protein hits (a measure of false discovery rate). **Score**: The score is calculated by MaxQuant using the formula: *Sum PEP Score = $\sum_i -log_{10}(PEP_{best\ peptide,i})$*, where PEP = Posterior Error Probability. **Identification type CON1**: Whether this protein group was identified in the control (untreated) by MS/MS or by matching between runs (can increase identification chances). **Identification type SAM1**: Whether this protein group was identified in the sample (R18-treated) by MS/MS or by matching between runs (can increase identification chances). **Sequence coverage CON1 [%]**: Percentage of the lead protein sequence covered by peptides in the control (untreated). **Sequence coverage SAM1 [%]**: Percentage of the lead protein sequence covered by peptides in the sample (R18-treated). **Intensity**: The summed intensities for all peptides assigned to this group. **Intensity CON1**: The summed intensities for all peptides assigned to this group in the control (untreated). **Intensity SAM1**: The summed intensities for all peptides assigned to this group in the sample (R18-treated). **LFQ intensity CON1**: The normalised intensity for all peptides assigned to this group in the control (untreated). **LFQ intensity SAM1**: The normalised intensity for all peptides assigned to this group in the sample (R18-treated). **MS/MS Count CON1**: Peptide spectrum matches in the control (untreated). **MS/MS Count SAM1**: Peptide spectrum matches in the sample (R18-treated). **MS/MS Count**: Peptide spectrum matches in the experiment. **Only identified by site**: If "+", a modification site alone was used to identify this protein. **Reverse**: If "+", this group is likely a false discovery, identified using the reverse decoy database. **Potential contaminant**: If "+", this group is a likely contaminant. **id**: A unique ID for this protein group. **Peptide IDs**: The IDs of the peptides assigned to this group. **Peptide is razor**: Shows which peptides are shared with other groups. **Mod. peptide IDs**: ID to look up these peptides in associated modifications file. **Evidence IDs**: ID to look up these peptides in associated evidence file. **MS/MS IDs**: ID to look up these peptides in associated MS/MS summary file. **Best MS/MS**: IDs of the peptides with the best matching MS/MS spectra for this group. **Oxidation (M) site IDs**: IDs to look up these peptides in the associated Oxidation data file. **Oxidation (M) site positions**: Positions of oxidation sites in the lead protein of this group. <u>**Sheet 3: LFQ**</u>. A summary of proteins identified in the experiment, after filtering. **Accession No:** The Uniprot Accession ID for this protein. **Peptides CON1 (CON2, C5, etc):** The number of peptides assigned to this protein in the control (untreated). **Peptides SAM1 (SAM 2, S5, etc):** The number of peptides assigned to this protein in the sample (R18-treated). **Mol. weight [kDa]:**

The molecular weight of the protein. **Score:** The score calculated by MaxQuant using the formula: *Sum PEP Score* = $\sum_i -log_{10}(PEP_{best\ peptide,i})$, where PEP = Posterior Error Probability. **LFQ intensity CON1:** The normalised intensity for all peptides assigned to this group in the control (untreated). **LFQ intensity SAM1:** The normalised intensity for all peptides assigned to this group in the sample (R18-treated). **Protein Name:** The Uniprot fasta header(s) for this protein group
(XLSX)

**S3 Table. Raw mass spectrometry intensity data for replicate 2.**
(XLSX)

**S4 Table. Raw mass spectrometry intensity data for replicate 5.**
(XLSX)

**S5 Table. Raw mass spectrometry intensity data for replicate 6.**
(XLSX)

**S6 Table. Raw mass spectrometry intensity data for replicate 7.**
(XLSX)

**S7 Table. Raw mass spectrometry intensity data for replicate 8.**
(XLSX)

**S8 Table. Amino-acid composition around predicted 14-3-3 binding sites.** The raw numbers of each amino acid at each position around predicted 14-3-3 binding sites, which are used to make Fig 2C. Position 7 in the table is the phosphorylation site essential for 14-3-3 binding, which corresponds to position 0 in the Fig 2C. Group 2, 3 and 4 correspond the top 20–40%, 40–60% and 60–80%, respectively.
(XLSX)

**S9 Table. Quantification of pulldown experiments.** Quantification and statistical analysis of the band intensities from triplicated experiments in Fig 3D–3F.
(XLSX)

**S10 Table. Quantification of GFP-Borr signal.** Quantification of GFP-Borealin signal, on which Fig 4B is based on.
(XLSX)

**S11 Table. Frequencies of bi-oriented homologous centromeres of chromosome 3.** The numbers of oocytes showing bi-oriented or mono-oriented homologous centromeres of chromosome 3, on which Fig 4E is based on.
(XLSX)

**S12 Table. Parameters used for mass-spectrometry.**
(XLSX)

## Acknowledgments

We are grateful to the members of the Ohkura laboratory for their help and for their contributions and for generating reagents, and especially to Emma Peat for her technical assistance. The Bloomington Drosophila Stock Center/Resource Center (National Institutes of Health grants P40OD018537 and 2P40OD010949-10A1) and the Transgenic RNAi Project at Harvard Medical School (National Institutes of Health/National Institute of General Medical Sciences grant R01-GM084947) provided fly stocks and reagents.

## Author Contributions

**Conceptualization:** Charlotte Repton.

**Formal analysis:** Charlotte Repton, Christos Spanos, Hiroyuki Ohkura.

**Funding acquisition:** Juri Rappsilber, Hiroyuki Ohkura.

**Investigation:** Charlotte Repton, C. Fiona Cullen, Mariana F. A. Costa, Christos Spanos, Hiroyuki Ohkura.

**Supervision:** Juri Rappsilber, Hiroyuki Ohkura.

**Writing – original draft:** Charlotte Repton, Hiroyuki Ohkura.

**Writing – review & editing:** Charlotte Repton, Hiroyuki Ohkura.

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
