## [Decision Letter · Decision Letter 0]

14 Jan 2022

Dear Dr Ohkura,

Thank you very much for submitting your Research Article entitled 'The phospho-docking protein 14-3-3 regulates microtubule-associated proteins in oocytes including the chromosomal passenger Borealin' to PLOS Genetics.

The manuscript was fully evaluated at the editorial level and by independent peer reviewers. The reviewers appreciated the attention to an important problem, but raised some substantial concerns about the current manuscript. Based on the reviews, we will not be able to accept this version of the manuscript, but we would be willing to review a much-revised version. We cannot, of course, promise publication at that time.

All three reviewers were enthusiastic about this work, and in balance, the result is somewhere between a major and minor revision. There are a quite a few comments related to making the paper more accessible by defining terminology or improving the organization. There is also some requests for more quantification and questions regarding the regulation of CPC centromere localization, and the function of S161 and S163. These comments can mostly be addressed by changes to the writing. In your rebuttal, however, also please address the requests for more experiments.

If you decide to revise the manuscript for further consideration at PLOS Genetics, please aim to resubmit within the next 60 days, unless it will take extra time to address the concerns of the reviewers, in which case we would appreciate an expected resubmission date by email to plosgenetics@plos.org.

[LINK]

We are sorry that we cannot be more positive about your manuscript at this stage. Please do not hesitate to contact us if you have any concerns or questions.

Yours sincerely,

Kim S. McKim

Guest Editor

PLOS Genetics

Gregory P. Copenhaver

Editor-in-Chief

PLOS Genetics

Reviewer's Responses to Questions

**Comments to the Authors:**

Reviewer #1: This manuscript by Repton, et.al. addresses the question of how spindles form in oocytes lacking centrosomes, specifically focusing on how the phospho-docking protein 14-3-3 regulates this process. A previous study from this lab demonstrated that 14-3-3 suppresses microtubule binding of Ncd away from chromosomes, to promote acentrosomal spindle assembly. This motivated the authors to search for additional proteins regulated by 14-3-3 in oocytes. They used a mass spectrometry approach to identify proteins whose binding to microtubules is increased/decreased by 14-3-3. The authors went on to study one of these newly identified factors in more detail, demonstrating that 14-3-3 binds to Borealin (a subunit of the CPC), and that two phosphorylations on Borealin regulate this binding. The authors then went on to mutate these sites in vivo, revealing effects on Borealin localization and centromere bi-orientation.

This study addresses an important problem and reports interesting findings that advance the field. The experiments are generally well executed, and the list of 14-3-3 regulated proteins generated will be a good resource for the field. Moreover, the work on Borealin represents a nice combination of in vitro and in vivo work. Despite these strengths, I have some suggestions that I think will improve the paper. Many of these are aimed at making the manuscript accessible to the broad readership of PLoS Genetics. While I think that the readers of this journal will find the topic of interest, the addition of more background information and some reorganization of the supplementary information will make it easier for a broad audience to understand (details below). I think that the below suggestions will improve this interesting paper.

Specific points

1. There is one experiment that I think could better connect the in vitro analysis in Figure 3 with the in vivo analysis in Figure 4. The authors note in the discussion (lines 336-338) that the non-phosphorylatable mutant they analyze in vivo is predicted to abolish 14-3-3 binding. However, this is not directly demonstrated in the manuscript (and doing so would better support their model). I suggest that they use the Borealin S163A (non-phoshorylatable) mutant in the pull-down experiments (same experiment as Figure 3D, but instead of WT Borealin (113-221), use this same fragment but with S163A).

2. The mass spectrometry data is likely to be a useful resource for the field and is a strength of the paper. Because of this, I think it would be helpful for the authors to better organize and present these data, so that readers can more easily analyze it. For example, I was not familiar with some of the terms in the supplementary tables (“LFQ intensity C6”, “LFQ.intensity.SAM”, “logSAM”, how the “score” was determined, etc.) but I could not find legends for these tables, so it was difficult for me to find this information. Also, I couldn’t figure out why the proteins were listed in the order they were – they didn’t appear to be sorted by accession number, or score, or number of peptides…I think it would be helpful to order/sort the lists of proteins in some rational way (maybe by which hits have the most confidence? Or by accession number?), and then provide legends to explain what is displayed in the tables and how the proteins are ordered.

3. Similar to comment #2 above, Table S1 in particular reports data that the field will be interested in. However, I think that it could be better organized. If I understand correctly, this table lists the proteins shown in Figure 1B; some of these proteins increase their binding to microtubules upon 14-3-3 inhibition (right side of the volcano plot), while others decrease (left side). It might be helpful if Table S1 was organized so that readers could easily tell which proteins increased/decreased/stayed the same. This information was provided for each protein in the table, but sorting the list somehow to reflect this value (maybe by ratio?) might help. Finally, I think it might be nice for the authors to make a separate table listing only the 47 proteins highlighted in the results section (lines 170-172); the authors point out that these are the hits with the most confidence, so I think readers might appreciate having a separate table listing only these proteins (rather than searching for these proteins in the larger list in Table S1).

4. The legend for Figure 1 has an error (it says there is a protein gel in Figure 1B, but this is not the case).

5. I was confused by one aspect of the model presented in the discussion, and think that it would be useful for the authors to expand upon and clarify their model so that the readers better understand the take-home message of the paper (perhaps they could also consider adding a model figure?). The authors propose that 14-3-3/Aurora B regulate the ability of Borealin to bind to spindle microtubules (lines 349-350) and that this regulation is important for biorientation (lines 350-351). However, in the presented experiments, it looks like the S163A mutant has reduced binding to both microtubules and centromeres – I don’t see the centromere dots in Figure 4A, and the fluorescence quantified in 4B is the total fluorescence and does not distinguish between microtubule and centromere populations. Given this, I think it possible that 14-3-3 may also regulate the ability of the CPC to bind centromeres. If this is the case, then could it be that delocalization of the CPC from centromeres is what causes the biorientation defects? If so, I think the authors should discuss this possibility – a broader discussion how the authors envision the regulation working would enhance the manuscript.

6. It would be helpful for the reader if there was more background information on Drosophila 14-3-3 proteins. For example, it is noted that there are two isoforms (line 91-92), and that 14-3-3n depletion disrupts spindle assembly (line 107), but more information would be helpful. What is the other isoform called? Have people tried to deplete it, and if so, was there a phenotype? In the authors’ previous study on NCD, did they assess both isoforms? Adding this type of broader background information would help people not familiar with the 14-3-3 literature better understand the current study. Additionally, in the results section the authors mention an inhibitor that they predict inhibits both isoforms. Having more information about the isoforms (e.g. how similar they are) and about the inhibitor would help the reader evaluate this claim.

7. I think it would be helpful to quantify the intensity of the bands on the gels in Figure 3 to give a more quantitative view of how much protein is pulled down. For example, the right-most band in the GST-pull down gel in figure 3D (PKD2 minus) looks a lot weaker than the rightmost band in the GST-pulldown gel in figure 3E (PKD2 minus/AurB minus), even though these are similar experimental conditions. This therefore made me wonder how much the results vary between experiments, so I think doing some sort of quantification of band intensity could help the readers understand the results better. It could be that different amounts were loaded on the gel in the different experiments (the PKD+ band in Figure 3D also looks lighter than the PKD+ band in Figure 3E, so maybe the relative amounts of protein that are pulled down with and without PKD2 are the same in the two experiments); quantifying these experiments would therefore help the reader better interpret the results.

8. This reviewer was left wondering which kinase phosphorylates S163, to mediate the regulation demonstrated in the manuscript. I am not suggesting that the authors do experiments to figure this out (I think that is beyond the score of the manuscript), but I think it would be nice if the authors could speculate on this in their discussion. They use PKD2 in their in vitro experiments, because it had been used previously to phosphorylate a similar site in NCD. But is this the kinase that they think phosphorylates this site in vivo? If not, what are some candidates? This suggestion is optional, but since I think that some other readers might have the same question, I think it would be a nice addition to the paper.

Typos/small wording changes:

- Line 34: “binding” should be changed to “bind”

- Line 66: should be “limit spindle assembly”

- Line 140: should be “ability to bind microtubules”

- Line 148: should be “through a sucrose cushion”

- Line 194: “presence of predicted”

- Line 300: “that are bioriented” (not which)

- Line 302: “Borealin RNAi oocytes” (instead of “RNAi oocytes”)

Reviewer #2: In this manuscript, Repton and colleagues report the identification of a role for 14-3-3e in the regulation of the Chromosomal Passenger Complex (CPC) in meiosis. They show that 14-3-3e binds the CPC subunit Borealin, thereby preventing its interaction with microtubules (MTs). This discovery was enabled by the authors’ development of a new assay for the identification of MT-binding proteins in ovary extracts, which I found impressive. Using 14-3-3 chemical inhibitors, they identified several proteins whose binding to MTs is blocked by 14-3-3. In this way, Ncd was recovered, providing a validation of previous findings by the same group. Borealin was also one of the hits, which they went on to study.

This study makes for a short and compact manuscript that presents few experimental results. Nevertheless, these results are quite novel and significant for the fields of molecular regulation of meiosis and potentially also mitosis. The experiments are sound and generally well controlled, the conclusions are justified by the results, the figures are well presented and the paper is very clearly written. I will be very supportive of publication after consideration of the following points:

1- Do failures in the mechanisms proposed affect egg viability? In the rescue experiments shown in Fig 4D-E, if chromosomes mis-segregate in meiosis, resulting aneuploid embryos may fail to hatch. Only chromosome 3 is scored for bi-orientation. If chromosomes X and 2 were also scored, the percentage of eggs with at least one misoriented chromosome could be very high. However, I don’t know if this experiment would work because the maternal driver used (BDSC 7063) is so strong that it causes a large fraction of embryos to abort, at least in our hands.

2- In Fig 4A-B, the total intensity of GFP-Borr at the spindle + centromeres was measured. However, the mechanism as proposed affects the ability of Borr to bind MTs only. It would be interesting to test if the same mechanism also affects the ability of Borr to localize to centromeres. Could the authors measure the intensity at centromeres only with their existing data?

3- The Western shown in Fig 4C is not clear enough. I think it should be redone. It is important to know that expression levels of the 3 forms of GFP-Borr are equal or very similar. Otherwise, the differences in fluorescence intensity measured could be due to different expression levels.

4- The authors propose that 14-3-3 binding of Borr prevents Borr from binding MTs away from the meiotic spindle. However, this is not tested experimentally. It would be good to test it with the GFP-Borr transgenic flies. To this end, could the authors measure the fluorescence intensity of their different variants of GFP-Borr on MTs of the polar body?

5- In all biochemical experiments, 14-3-3e (epsilon) is used and never 14-3-3z (zeta). Do the authors assume that 14-3-3z would behave in the similar way? This should at least be discussed if not tested. Also, the name 14-3-3e should be used throughout when referring to the protein used in experiments.

6- Results in Fig 3E suggest that 14-3-3e may bind Borr to some degree when unphosphorylated. Would the authors like to comment on it? It looks like this basal binding may be also inhibited by phosphorylation by AurB.

7- Do the authors think that PDK2 may be a kinase that contributes to phosphorylate S163 in vivo? Can they speculate on what other kinases may be involved?

8- I wonder if the regulation of Borr/CPC by 14-3-3 plays a role in cytokinesis. We could imagine that 14-3-3 binding helps prevent CPC binding to the MTs until cytokinesis, when it is recruited to the central spindle. A phosphatase could dephosphorylate S163 during mitotic exit to promote this transition. This may be an idea for future experiments and/or a discussion point.

9- Fig 1B mentioned in the legend (Protein gel) is missing.

Reviewer #3: "The phospho-docking protein 14-3-3 regulates microtubule-associated proteins in oocytes including the chromosomal passenger Borealin" (PGENETICS-D-21-01648) by Prof. Hiroyuki Ohkura and colleagues for PLOS Genetics.

In this paper, the authors used a differential screening approach to identify MAPs regulated by 14-3-3 proteins and required for meiotic spindle assembly. They isolated MAPs and other proteins that show higher or lower affinity for microtubules in the presence of the 14-3-3 inhibitor (R18).

Among isolated proteins they started to study Borealin, a subunit of the Chromosome Passenger Complex. This protein displays a putative phosphorylation site (S163) that matches a 14-3-3 binding site. An Aurora B phosphorylation site is found nearby (S161). This motif is similar to the motif present in the Ncd kinesin. Binding of 14-3-3 to Ncd is regulated by PKD2 and Aurora B Phosphorylation. This team has previously shown an interesting regulatory event for meiotic spindle assembly. PKD2 phosphorylates Ncd on a 14-3-3 binding site. Following phosphorylation, 14-3-3 binds to Ncd that is released from microtubules. However, phosphorylation by Aurora B triggers 14-3-3 release and local binding of Ncd on the meiotic spindle.

They show that the disorder region of Borealin binds to 14-3-3 after adding PKD2. This interaction is inhibited after Aurora B addition. Indicating that this region is likely regulated by both Kinases.

The also show that S161 and S163 are required for centromere and for chromosome bi-orientation.

Main concerns.

I appreciate the differential screening strategy used to isolate proteins (in particular MAPs) sensitive to inhibition by 14-3-3. Riding the wave of their first publication on Ncd, regulated by 14-3-3 and Aurora B, they analyzed the in vitro and in vivo effects of an almost identical motif. Unfortunately, the link between the initial screen for MAPs and the final result on the centromeric de-localization of Borealin are unclear. Finally, several results are correlative and essential controls are missing. The mechanism is speculative.

1-It is essential to show that S161 and S163 of Borealin are essential for interaction with 14-3-3. This has not been shown properly in the in vitro experiments. A S163A mutant should not be able to interact with 14-3-3 in the presence of PKD2. Similarly a S161A mutant should maintain interaction with 14-3-3 in the presence of Aurora B and PKD2. This is essential to connect in vitro and in vivo observations.

Other important concerns that nevertheless need to be amended.

1-This is a matter of taste but I believe too much emphasis is made on the previous Ncd publication. The mechanism involved may be completely different for Borealin.

2- Would it be possible to also include in figure 2 the identity of the proteins that are less present in the MT pellet in the presence of the R18 inhibitor.

3) L611. Where is the gel in the figure 2.

4) RCC1 levels are not comparable and significantly different (there is a discrepancy between Lane 283 and Figure 4B lower panel). Is it because of the image acquisition? Is it because the S161 mutant triggers loss of RCC1 targeting?

5) The images of IF acquisitions are very small and it is difficult to see the differences. Scale bar are missing in some panels.

6) Western Blots are not of optimal quality. Load same amounts of proteins (Figure 4C)

7) Figure 3. Please provide quantitation, I do not see differences in Fig 3D between ++ and +. Alternatively, use one condition.

8) Figure 3 E. There is more MBP-Bor bound to GST-14-3-3 in the presence of PkD2 (well ° 5) but there is obviously more GST14-3-3 on the beads. See previous comment (well 8) There seems to be more bound MBP-Bor to GST-13-3-3 without any of the 2 Kinases, how it is possible?

9) What is the evidence that regulation of interactions between 14-3-3 and Borealin (by PKD2 and Aur B) is regulated via Ser161 and S163. Are these sites phosphorylated in vivo? (See my main concern N°1).

10-There is also a 14-3-3 binding site in INCENP. Please comment for potential combinational effects on CPC localization.

11-L667. Arrowheads are not visible on the figure.

12-In a previous study this team has shown that the CPC regulates MT nucleation on the meiotic spindle. However, MTs appear to be well nucleated in the absence of Borealin that is a member of the CPC complex. This is should be clarified.

13-Usually, the RNAi/Rescue system is validated by Western Blot to challenge depletion of the endogenous protein and persistence of the rescuing tagged protein. This is not shown in this study. Please provide evidence the system is working. Is there a Borealin antibody to validate the strategy?

14-There is confusion between spindle and centromere localization. The screen aimed to find proteins for which 14-3-3 binding abolishes interaction with meiotic MTs. However the Borealin protein (and possibly other CPC subunits) is located on the centromeres and not on the meiotic spindle.

15-Lane 365-366-“ Our study identified a novel mechanism  regulating microtubule binding activity of the CPC, which is essential for bi-orienting  homologous chromosomes in oocytes.   » There are no direct evidences that MT binding activity of Bor is involved in CPC centromere localization (S161 and S163 appears important). This is over interpretation.

**Have all data underlying the figures and results presented in the manuscript been provided?**

Reviewer #1: Yes

Reviewer #2: Yes

Reviewer #3: Yes

PLOS authors have the option to publish the peer review history of their article (what does this mean?). If published, this will include your full peer review and any attached files.

Reviewer #1: No

Reviewer #2: No

Reviewer #3: No

---

## [Decision Letter · Decision Letter 1]

4 Apr 2022

Dear Dr Ohkura,

Thank you very much for submitting your Research Article entitled 'The phospho-docking protein 14-3-3 regulates microtubule-associated proteins in oocytes including the chromosomal passenger Borealin' to PLOS Genetics.

The manuscript was fully evaluated at the editorial level and by independent peer reviewers. The reviewers appreciated the attention to an important topic but identified some concerns that we ask you address in a revised manuscript

We therefore ask you to modify the manuscript according to the review recommendations. Your revisions should address the specific points made by each reviewer.

[LINK]

Yours sincerely,

Kim S. McKim

Guest Editor

PLOS Genetics

Gregory P. Copenhaver

Editor-in-Chief

PLOS Genetics

Two of the three reviewers had no or only minor comments. One reviewer requested an additional experiment with the S161A mutation. While it would be an improvement to have this data, I understand that time and technical constraints may make this difficult to accomplish. In addition, the S163A experiment does make the paper stronger. I would like you to consider doing this experiment, but if not, state explicitly (like on pg 11) that you have not shown S161 regulates interactions with 14-3-3 in an AuroraB-dependent manner. I would also like you to emphasize on pg 12 that the Borealin RNAi is probably a partial knockdown, because its phenotype is much milder that the loss of other CPC components. With these changes and the others asked for by two reviewers, I don't think sending the manuscript out for review will be necessary.

Reviewer's Responses to Questions

**Comments to the Authors:**

Reviewer #1: The manuscript by Repton et.al. is substantially improved and the authors did a good job addressing my concerns. The manuscript is now more accessible to a broad audience, and I think will be of interest to the readers of PLOS Genetics.

I found one small typo but otherwise have no further suggestions:

Line 204: “which increases or decreases” should be “that increase or decrease”

Reviewer #2: My points have been addressed in an acceptable manner. I am in favor of accepting the revised manuscript for publication.

Reviewer #3: I had requested a number of additional experiments. Despite the changes in the manuscript, I note that the authors did not perform any of these experiments. This is detrimental to the quality of the manuscript and I think that the in vitro experiment with the S161A mutant would have really provided the missing piece of the puzzle to make a solid story.

I believe that the regulation of mitotic proteins, including CPC by 14-3-3 proteins is an important finding in the field of cell division. However, while I appreciate the changes in the writing that have improved the manuscript, I still have serious concerns because none of the experiments I requested have been performed.

It is really unfortunate that the authors ignored my comment on serine 161. Even if the effect of Aurora B on 14-3-3 is “likely” not due, (according to them,) to another phosphorylation this is not demonstrated. It is all the more unfortunate that the most complicated experiment (in vivo study) was performed with the variant harbouring the S161A mutation. Given the team expertise expertise, I find it hard to believe that the time constraint is the first reason given for not doing this experiment (request for extension period over the 2 months period is usually given by the editors). In the end, it is rather unfortunate for the overall quality of this manuscript because Figure 3 shows the importance of Aurora B in vitro and Figure 4 shows the importance of Ser161 in vivo. There is a real gap here that could have been filled by doing the requested experiment. I suggest doing this experiment, and correcting their protein loading problems in the gels.

Other minor comments (version with annotated changes).

If the experiment in Fig 3 was performed 3 times, there should be some quantification with appropriate tests (n=3, P<, SD…etc).

. “S2 Fig). Among the 47  proteins that increased microtubule binding in the presence of the 14-3-3 inhibitor, we found 48 known protein-protein interactions among these 47 proteins, which is significantly higher than  expected from a random set of 47 proteins (17 interactions; p=1.2e-09) » How this is possible ?

. Lane 308. The authors are entangled with their signal quantifications, (Western Blots and IF images). Please clarify. The authors claim The RCC1 levels are not different (this is not reflected by the statistical tests). I understand that making new experiments at this stage using another DNA probe is complicated. I would suggest to use RCC1 as a probe to position the meiotic spindle and to stick on GFP signal quantification. However, this is possible only if the proteins are expressed are similar levels and if the GFP blot is OK. As I mentioned in my previous review the anti-GFP Western blot is far from optimal, I suggest using another GFP antibody, (or performing IP using GFP TRAP beads) and load similar amounts of proteins. It is not an unjustified request to ask for quality Western blots for a high-quality journal like PLOS Genetics.

. Lane 409-411. As our proposed mechanism can provide a self amplification  loop, in theory this could confer a localised switch-like property to CPC activity during  cytokinesis and/or help to accumulate the CPC to a tighter region.  

I fail to understand the self-amplification loop mechanism, please explain.

**Have all data underlying the figures and results presented in the manuscript been provided?**

Reviewer #1: Yes

Reviewer #2: Yes

Reviewer #3: Yes

PLOS authors have the option to publish the peer review history of their article (what does this mean?). If published, this will include your full peer review and any attached files.

Reviewer #1: No

Reviewer #2: No

Reviewer #3: No

---

## [Editor Report · Decision Letter 2]

27 Apr 2022

Dear Dr Ohkura,

We are pleased to inform you that your manuscript entitled "The phospho-docking protein 14-3-3 regulates microtubule-associated proteins in oocytes including the chromosomal passenger Borealin" has been editorially accepted for publication in PLOS Genetics. Congratulations!

Yours sincerely,

Kim S. McKim

Guest Editor

PLOS Genetics

Gregory P. Copenhaver

Editor-in-Chief

PLOS Genetics

Comments from the reviewers (if applicable):

**Data Deposition**

http://datadryad.org/submit?journalID=pgenetics&manu=PGENETICS-D-21-01648R2

**Press Queries**

---

## [Editor Report · Acceptance letter]

31 May 2022

PGENETICS-D-21-01648R2 

The phospho-docking protein 14-3-3 regulates microtubule-associated proteins in oocytes including the chromosomal passenger Borealin 

Dear Dr Ohkura, 

We are pleased to inform you that your manuscript entitled "The phospho-docking protein 14-3-3 regulates microtubule-associated proteins in oocytes including the chromosomal passenger Borealin" has been formally accepted for publication in PLOS Genetics! Your manuscript is now with our production department and you will be notified of the publication date in due course.

With kind regards,

Anita Estes

PLOS Genetics

On behalf of:
